# An Emerging Animal Model for Querying the Role of Whole Genome Duplication in Development, Evolution, and Disease

**DOI:** 10.3390/jdb11020026

**Published:** 2023-06-06

**Authors:** Mara Schvarzstein, Fatema Alam, Muhammad Toure, Judith L. Yanowitz

**Affiliations:** 1Biology Department, Brooklyn College at the City University of New York, Brooklyn, NY 11210, USA; 2Biology Department, The Graduate Center at the City University of New York, New York, NY 10016, USA; 3Biochemistry Department, The Graduate Center at the City University of New York, New York, NY 10016, USA; 4Magee-Womens Research Institute, Pittsburgh, PA 15213, USA; yanowitzjl@mwri.magee.edu; 5Department of Obstetrics, Gynecology, and Reproductive Sciences, University of Pittsburgh School of Medicine, Pittsburgh, PA 15213, USA

**Keywords:** whole genome duplication, polyploidization, tetraploidy, polyploidy, aneuploidy, *Caenorhabditis*, adaptation, stress resistance, carcinogenesis, allometric relationships

## Abstract

Whole genome duplication (WGD) or polyploidization can occur at the cellular, tissue, and organismal levels. At the cellular level, tetraploidization has been proposed as a driver of aneuploidy and genome instability and correlates strongly with cancer progression, metastasis, and the development of drug resistance. WGD is also a key developmental strategy for regulating cell size, metabolism, and cellular function. In specific tissues, WGD is involved in normal development (e.g., organogenesis), tissue homeostasis, wound healing, and regeneration. At the organismal level, WGD propels evolutionary processes such as adaptation, speciation, and crop domestication. An essential strategy to further our understanding of the mechanisms promoting WGD and its effects is to compare isogenic strains that differ only in their ploidy. *Caenorhabditis elegans* (*C. elegans*) is emerging as an animal model for these comparisons, in part because relatively stable and fertile tetraploid strains can be produced rapidly from nearly any diploid strain. Here, we review the use of Caenorhabditis polyploids as tools to understand important developmental processes (e.g., sex determination, dosage compensation, and allometric relationships) and cellular processes (e.g., cell cycle regulation and chromosome dynamics during meiosis). We also discuss how the unique characteristics of the *C. elegans* WGD model will enable significant advances in our understanding of the mechanisms of polyploidization and its role in development and disease.

## 1. Whole Genome Duplication in Development, Evolution, and Disease

Whole genome duplication (WGD) or polyploidization is vital for normal development [1], tissue homeostasis [2,3], and regeneration [4] (Figure 1). It is also a driver of evolutionary processes [5] including adaptation [6], speciation [7], and crop domestication [8]. In addition, WGD can drive pathological conditions, especially oncogenesis [1,9,10]. In nature, WGD exists at the cellular, tissue, organ, and organismal levels. Polyploid cells are normally found in most multicellular diploid organisms [11,12]. In humans, certain cells from the placenta [13], mammary glands [14], liver [15], heart [16], skin [17], and bone marrow must become polyploid to perform specialized functions [18], for tissue homeostasis [3], or as part of the wound healing process [19]. For instance, during development, megakaryocytes acquire 16 copies of the genome and thus become giant cells that are hypermetabolic. This polyploidization is a basic requirement to produce platelets in the blood, as platelets are produced by fragmenting the cytoplasm of megakaryocytes [20,21,22]. In addition, WGD is involved in oncogenesis, metastasis, and the development of resistance to chemotherapeutic drugs, and polyploid cells may accumulate in patients with neurodegenerative disorders, cardiovascular disease, and diabetes [23,24,25,26].

Polyploid cells can arise by cell fusion resulting in larger multinucleate cells, by endoduplication (alternating cycles between G and S phases without undergoing mitosis or cytokinesis) resulting in mononucleated cells, or by endomitosis (the cell enters mitosis but does not complete cell division) to generate either mononucleated giant cells or multinucleated cells [27]. In some cyanobacteria, polyploidy is established and maintained by misregulation of the replisome machinery [28,29,30]. 

### 1.1. Causes and Downstream Effects of WGD

The outcomes of WGD are diverse in nature—ranging from changes in gene expression to altered cell size and scaling and to adaptation to stress [31,32] (Figure 2). The effects of polyploidization are surprisingly similar in different biological contexts (e.g., in cells, tissues, and organisms) [4,11,33,34] and regardless of the biological process (e.g., during development, evolution, regeneration, and oncogenesis) [1,3,35,36,37,38,39,40,41] where WGD takes place (Figure 2). A common feature of WGD is that it is induced by stress, and it provides the changes that result in plasticity and thus enhances adaptation to stress in normal and pathological conditions. For instance, environmental stresses induce WGD in tissues (e.g., wounding or viral infection) [42,43,44] or organisms (e.g., large changes in temperature or water availability and salinity) [45,46,47,48,49,50]. Furthermore, stress-induced WGD is a platform for genomic diversity and versatility that provides the means for rapid adaptability (Figure 1 and Figure 2). Whereas in whole organisms, this can elicit a competitive advantage in a changing environment [32,51]; in cancer, it elicits an adaptive advantage that promotes oncogenesis, metastasis, and the development of drug resistance [32,51]. The latter is similar to how the pathogenic yeast *Candida abdicans* becomes resistant to antifungal drugs [6]. Therefore, it is widely expected that queries about mechanisms that give rise to WGD and its effects in both sub-organismal and organismal systems will likely reveal additional commonalities across systems and processes. 

Polyploidy is often confused with aneuploidy, in part because, in humans, both phenomena are hallmarks of cancer [1,39,40,52]. Observations from numerous studies identify WGD in about 30% of human cancers [53,54] and as a precursor for many malignancies in cancer evolution [39,55,56,57,58,59]. Although WGD does not seem to require a preexisting cancer driver mutation; it most often emerges after oncogenic mutations, such as lesions in the *TP53* genes [53,56]. WGD promotes diversification of copy number alterations (CNAs) and is permissive for other chromosomal aberrations and genome instability associated with poor cancer patient prognosis [39,58,59]. Interestingly, a recent study revealed a mechanism by which WGD enhances cancer-promoting alterations in DNA organization and gene expression [58]. Induction of WGD can reduce the levels of proteins involved in chromatin packaging, resulting in reorganization of the 3D DNA compartments and domains. Over time, loss of this domain structure predisposes the cell to additional cancer-promoting alterations and expression of oncogenic genes [58]. Polyploidization can also be protective, as in the liver where it both provides the basis for genetic adaptation to hepatotoxic stress and also protects from malignant transformations [21,60].

Organismal WGD is common in extant plants [7,32,61,62,63] and is also observed in protists [64], fungi [51,65,66], bacteria, and archaebacteria [67,68] and in several animal clades [2,12,35,36,40,41,69,70]. In both prokaryotes and eukaryotes, the effects of WGD are comparable with increased resistance to UV irradiation, cell size, adaptability to environmental changes, and evolvability [28,71]. Evidence for ancestral WGD events is found in the genomes of organisms of most clades, including vertebrates [36,72,73]. Polyploid metazoans arise due to errors during meiosis that result in the formation of diploid gametes that when fertilized give rise to polyploid organisms. Fertilization between diploid (2n) and haploid (n) gametes results in a triploid (3n) organism; fertilization between two diploid (2n) gametes results in a tetraploid (4n) [33,35]. When the diploid gametes of two related species fuse, the resulting organism is said to be an *allopolyploid*, whereas when diploid gametes from the same species fuse the result is said to be an *autopolyploid* [73]. Polyploidization from the cell to the organismal level most often results in lethality [32,74]. For instance, human triploid and tetraploid embryos die during gestation or soon after delivery, but triploid and tetraploid mosaic children may survive after birth [75,76]. This lethality may be caused by errors partitioning the additional chromosomes during cell division, by WGD-driven increases in cell size that interfere with tissues or organ scaling, and by epigenetic changes and gene expression changes that affect dose-dependent processes and complexes [73]. The latter may be especially true in human triploids and tetraploids which may have difficulty equalizing expression between the silenced X chromosomes and the extra sets of autosomes since variable numbers of Barr bodies may be found in these cells [75]. A newly formed polyploid may very occasionally overcome these obstacles, e.g., by establishing balanced genome expression. Then, if polyploidy provides a competitive advantage—as in a changing environment—polyploid lineages may outgrow their diploid counterparts and become established [32,41,73]. The established polyploid population will continue to evolve and then may “diploidize” to become a new species that has an adaptive advantage compared to the parental diploid species [32]. 

### 1.2. Laboratory Multicellular Organism Models for WGD

A major impediment to studying WGD and its impacts on the physiology and function of cells, tissues, and organs is the scarcity of laboratory models that permit comparison between isogenic organisms with differing ploidy [33]. Many laboratory models are sterile or embryonic lethal when polyploid [77,78,79]. Generation of autopolyploid laboratory model organisms that are not sterile (e.g., plants) frequently involves the use of chemicals such as colchicine, known to cause aneuploidy and genetic mutations. These polyploids, therefore, may not be truly isogenic with the parental diploid strains they were derived from [51,73]. The recently developed method for generating tetraploids in *Caenorhabditis* utilizes transient knock-down of a meiosis-specific cohesin (i.e., *rec-8*) by RNA interference to produce diploid gametes [80]. Therefore, the derived tetraploids are unlikely to differ significantly from the diploid strains from which they were derived (see Figure 3A for an overview of the method) [80]. 

## 2. Caenorhabditis Elegans as an Animal Laboratory Model for Understanding WGD

Several attributes of *C. elegans* nematodes make it an exceptional laboratory model for querying WGD: (1) it has a small number of chromosomes (five autosomes and a sex chromosome) allowing for easy visual evaluation of ploidy and differentiation between polyploidy and aneuploidy [81] (Figure 3B,C). (2) Despite *C. elegans* being a self-fertilizing hermaphroditic species, males can arise spontaneously in hermaphrodites by meiotic nondisjunction at a 0.2% frequency [82,83,84], allowing for genetic analysis [85]. Males are also produced by mating between hermaphrodites and males (frequency > 40%). (3) Each diploid hermaphrodite produces 200–300 offspring and has a lifecycle of three to four days from egg to egg, both of which facilitate multigenerational ecological and evolutionary studies. (4) *C. elegans* is normally a diploid but like vertebrates (including humans), it has polyploid tissues or cells [86]. These include the intestine (20 cells with a total of 30–34 nuclei and a ploidy of 32C (or 32 chromosome sets) per nucleus) and the hyp7 cell (a syncytium of 139 nuclei with an average ploidy of 10.7C) [86,87].

Critically, the quick and reliable method to generate fertile and viable tetraploid *Caenorhabditis* strains from nearly any diploid strain [80] allows direct comparison of identical cells in isogenic organisms with differing ploidy (Figure 3). The availability of hundreds of mutant strains that impinge on different aspects of development, aging, stress, and reproduction will facilitate the directed study of genes in many of the processes that polyploidy has been reported to affect. The transparency of the worms allows visualization and direct comparison of physiology and function in the live animal. Combined, these characteristics make the *C. elegans* WGD model unique and provide an unprecedented opportunity to answer long-standing questions about organismal polyploidization and its effects in a multicellular animal.

### 2.1. Polyploidy and Aneuploidy in C. elegans 

In humans, all monosomies, except for the X or Y chromosomes, are lethal, whereas trisomies 13, 18, 21, X, and Y are viable. Autosomal trisomies can cause severe developmental problems; sex chromosome trisomies have comparatively mild effects, presumably because of Barr body formation (extra X chromosomes) or the dearth of active genes (Y chromosome). In *C. elegans*, some aneuploidies are tolerated. These include X-chromosome trisomies (2A;3X = 13 chromosomes triplo-X hermaphrodites) and monosomies (2A;1X = 11 chromosomes XO males shown in Figure 4) [88]. Compared to diploid hermaphrodites, triplo-X animals are shorter and fatter (i.e., Dumpy phenotype), lower fertility, and slower growth rate [88]. Trisomy of chromosome 4 (LGIV) has also been reported [89]. Although animals with this autosomal trisomy are viable and have a general morphological appearance that seems normal, they produce 50% fewer progeny than normal hermaphrodites [90]. Interestingly, large chromosomal fragments, generally more than an Mb in size, have been generated and preserved as free duplications and have been found for part of each of the chromosomes of *C. elegans* [91].

In nature, most triploids are inherently unstable, unless they reproduce asexually [92]. For instance, the parasitic nematodes of the *Meloidogyne incognita* group (MIG) are stable hypotriploids that reproduce by asexual parthenogenesis [93]. *C. elegans* triploid animals (3A;3X and 3A;2X shown in Figure 4) can be generated by crossing diploid and tetraploid animals [94]. Only 15% of the embryos produced by triploid animals hatch [94], and those that do hatch tend to be diploids or become diploid in the next generation [95,96,97]. Triploid progeny inviability is likely because the odd number of homologs results in abnormal meiotic pairing and recombination [95,98,99], yielding aneuploid gametes that produce embryos that are inviable.

Victor Nigon reported the first induction of polyploid *C. elegans*—the first in any animal—in the laboratory in 1949 [96]. These tetraploid strains were generated by treating spermatogenic animals with either heat shock (several hours at 25 °C) or exposure to colchicine. Both treatments were followed by screening for ‘larger than normal’ animals and cytological assessment of chromosome numbers in diakinesis-stage oocytes, when the chromosomes are highly condensed prior to the meiotic divisions (e.g., in Figure 3C). Newly formed tetraploids produced a range of phenotypes in offspring that included: progeny with variable body sizes; infertile hermaphrodites and males; intersexuality; and progeny that had reverted to diploidy [96,97]. The frequency of these aberrant phenotypes decreased with continuous passaging but did not disappear, even after 20 generations of selection for tetraploids with high fecundity. Two major types of tetraploids were identified [94,96]: about 66% of the induced tetraploids produced very low frequencies of males (0.6%) compared to the ~0.2% incidence of males in diploids; the remainder produced an average of ~42% males. Interestingly, tetraploids of both classes could interconvert and give rise to a proportion of progeny of the other type, albeit at different frequencies [97]. After 30 generations, the patterns of inheritance of high and low incidence of male progeny did not change. The tetraploid strains that produced high proportions of males are now known to correspond to a 4A;3X genotype and those that produced a low incidence of males are known to correspond to a 4A;4X genotype (see Figure 4) [94]. Even in the most stable lines, tetraploid hermaphrodites had substantially fewer self-progeny (average of ~65 progeny) than diploids (250–300 offspring in wild type, N2) and reduced hatching (decrease of 13%) [94]. Nigon’s tetraploid strains were eventually all accidentally lost after 78 generations. Several other groups have isolated fertile and relatively stable tetraploid *C. elegans* and *C. briggsae* strains [80,94,99,100,101]. Like their predecessors, each of these tetraploid strains has been reported to produce reduced numbers of self-progeny and to sporadically diploidize. Since the diploids can rapidly overtake a population due to their higher fertility, *C. elegans* tetraploid strains need to be maintained by continuous selection for the largest animals—at least for the first tens of generations. Interestingly, an obligate tetraploid strain has been reported (CB3911) [100]. This strain arose spontaneously from a diploid carrying a mutation in the dosage compensation gene *dpy-27* which causes the lethality of XX diploids and XXXX tetraploids because of X-chromosome overexpression (Jonathan Hodgkin, personal communication). However, this strain produces viable *dpy-27*(*rh18*) 4A;3X hermaphrodites and 4A;2X males [100]. Studies of the early events post tetraploidization (e.g., in generations 1–10) therefore should shed light on how organisms acclimate to tetraploidy and become established. 

### 2.2. Polyploid and Aneuploid Animals Uncover C. elegans Modes of Sex Determination and Dosage Compensation

One of the most critical developmental decisions that a heterogametic organism must make early in life is whether to develop as male or female and, concomitantly, whether to activate dosage compensation. From studies in *C. elegans* with different ploidies, it became apparent that sex is assessed by the ratio of sex chromosomes to autosomes (X/A) and that induction of hermaphrodite fate activates dosage compensation, which is achieved by reducing expression from the pair of X chromosomes [83,102]. Male phenotypes arose from animals with 2A;1X, 3A:2X, and 4A;2X, whereas 2A;2X, 3A;3X, 4A;4X, and 4A;3X gave rise to hermaphrodite phenotypes [94] (Table 1 and Figure 4). 

Together, these studies led to the conclusion that an X:A ratio > 0.74 induces female/hermaphrodite identity and <0.67 promotes male development. Further support for these conclusions came from manipulation of the X/A ratio using X-linked gene duplications in the 2X;3A triploids to show at least three loci were involved in determining sex [101]. We now know the molecular details about the genes and processes involved both in assessing the X/A ratio and in executing male and hermaphrodite fates and dosage compensation [103]. Nevertheless, questions remain about how allotetraploids generated by the hybridization of different tetraploid *Caenorhabditis* species would assess the X/A ratio and whether manipulation or modification of this system could be used to stabilize polyploid organisms. 

## 3. Utilizing Polyploid *C. elegans* as Tools to Investigate Developmental Processes

In the last ~50 years, comparisons between *C. elegans* strains of differing ploidy have been utilized as tools to query developmental processes including meiosis and embryonic divisions, in addition to sex determination and dosage compensation. 

### 3.1. Understanding Early Events of Meiotic Prophase I

During meiosis, recombination between the duplicated maternal and paternal homologous chromosomes transiently connects the homologs to orderly partition them in the first division while maintaining sister chromatids together until the second division. To recombine, homologous chromosomes must first find one another in the nucleus (i.e., pair), become aligned along their lengths, and then stabilize their association by assembling the synaptonemal complex (i.e., synapse) [104]. Observations of the effects of having additional chromosomes in *C. elegans* triploid and tetraploid animals have helped to uncover rules underpinning homologous chromosome dynamics during meiosis [98,99]. Additional chromosomes did not significantly alter the initial steps of pairing into groups of three or four homologs in triploids and tetraploids, respectively. However, full alignment was often perturbed, and synapsis was frequently incomplete, resulting in the lengthening of the period in which synapsis normally takes place [98]. The imperfectly synapsed chromosomes became marked by di-methylation (H3K9me2), which has been proposed to prevent the synapsis defects from being detected by the quality control system [98]. Further analysis revealed that two pairs of synapsed homologous chromosomes form in tetraploids [99]. This pairwise synapsis also takes place in germ lines of mutants (i.e., *spo-11*) that are deficient in early steps of recombination, as in diploids. In triploid germ lines, one pair of homologs synapse, and the remaining homolog eventually folds onto itself, in a process known as self-synapse. In contrast to diploids, pairwise synapsis is disrupted in the presence of recombination-deficient mutations, suggesting that in *C. elegans*, as in other organisms, homologous recombination plays a role in promoting synapsis [99]. Polyploid animals heterozygous for translocation chromosomes (e.g., *nT1*(*IV*;*V*)) were utilized to test whether pairwise synapsis is dictated by full sequence homology along the chromosome length. Synapsis was normal in diploid translocation heterozygotes (i.e., *nT1/+* or *mln1/+*), as long as they contained the pairing center sequence that is required for homologous chromosomes to initially find each other in the nucleus [105]. In tetraploids (*nT1/nT1/+/+*), there was no preference for a synapsis partner as translocation chromosomes paired with normal chromosomes with the same frequency as with themselves. In contrast, associations and synapsis between pairs of homologous chromosomes were biased towards homologs with identical sequences in the triploids [99]. These findings support the proposed model in which the initiation of synapsis depends on pairing center-mediated associations of homologs, but the completion of synapsis along the homologous pair is promoted by recombination [99].

### 3.2. Understanding Meiotic and Early Embryonic Cell Divisions 

Female meiotic divisions are asymmetric. Each division gives rise to a large oocyte and a tiny polar body. The resulting egg contains a haploid set of chromosomes, as half of the chromosomes in each division are discarded into the polar bodies (Figure 5A,B). Therefore, *C. elegans* hermaphrodites trisomic for the X chromosome (2A;3X) would be expected to produce equal proportions of eggs that carry either one or two X chromosomes (one sister from the paired homologs +/− one sister from the self-synapsed chromosome). However, the observed X:XX ratio in these eggs is 2:1 [88]. Live and high-resolution visualization of the oocyte cell division in these trisomic worms showed that the self-synapsed chromosomes were preferentially partitioned into the polar body during oogenesis [95,106]. In addition, in triploid animals that have one set of homologous chromosome pairs (bivalents) and a set of unpaired chromosomes (univalents), the univalents were most often discarded into the polar body of the first oocyte division [106]. In these triploids, preferential segregation of the single sister chromatids into the polar body could also be observed in the second division. These results provide mechanistic insights into why triploid animals rapidly diploidize.

Several seminal studies have utilized tetraploid and haploid (1A;1X) *C. elegans* embryos to query cell division [107,108,109,110,111]. An overview of the relevant early developmental events, from fertilization to the first embryonic division, is shown in Figure 5.

**Figure 5 jdb-11-00026-f005:**
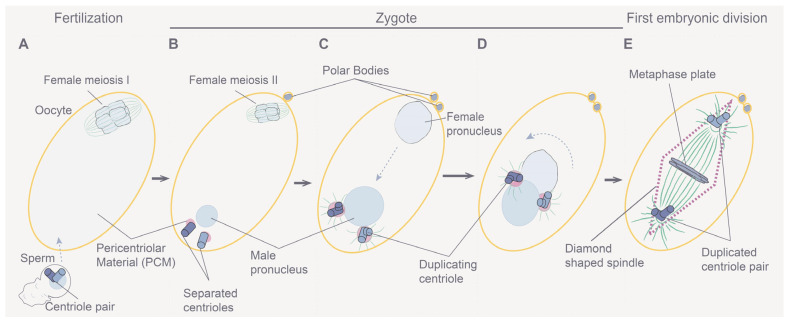
Early development from fertilization to the first embryo division. In addition to contributing a haploid genome, the sperm also contributes a pair of centrioles to the zygote upon fertilization (**A**). Fertilization by the sperm triggers the completion of the oocytes’ meiotic divisions, each of which generates a polar body containing half of the chromosomes from the division (**A**–**C**). The paired centrioles separate and remain in the proximity of the male pronucleus and, as they gather maternal pericentriolar material from the oocyte cytosol (**B**), they duplicate to form the first pair of centrosomes of the zygote (**C**–**E**) [112]. Concurrently, the male pronucleus grows as DNA decondenses (**B**,**C**), and the centrosomes then become attached to its surface (**C**) [113]. The female pronucleus migrates towards the male pronucleus as centriole duplication is completed (**C**,**D**). The duplicated centrioles form the poles of the first bipolar spindle of the embryo (**E**). The centrosomes and the metaphase plate make a diamond shape (dashed and red) that defines the length (centrosome–centrosome) and width of the spindle (longer axis of the metaphase plate at the equator) (**E**).

One study addressed how paternally inherited centrosomes become associated with the nucleus. Live imaging comparisons of zygotes with different ploidies suggested that centrosome attachment to the nuclear membrane depends on the size of the male pronucleus [107]. Diploid embryos with mutations in nuclear envelope genes that have aberrantly small male pronuclei initially had only one attached centrosome; the second centrosome only became attached as the male pronucleus surface increased [107]. This detached centrosome phenotype was suppressed in zygotes from tetraploid animals that had larger pronuclei. Conversely, the detached centrosome phenotype was enhanced in haploid zygotes with smaller pronuclei. It has been speculated that dynein motor accumulation on the pronuclear surface area is the limiting factor for centrosome attachment with a threshold of dynein required for both centrosomes to attach [109,114]. Of note, the haploid embryos in this study were generated utilizing conditional mutant sperm that lack nuclear DNA (i.e., *emb-27*(*g48ts*)) [109].

Another important study addressed whether the nuclear-to-cytosol ratio regulates cell cycle timing, a hypothesis that had been proposed from work in other organisms [115,116,117]. They compared diploid (N2 Bristol), tetraploid (strain SP343), and haploid embryos (generated by extracting the pronucleus from the fertilized egg before the zygote formed) [108]. In these embryos, different nuclear/cytosolic volume ratios were examined by extruding nuclei with little to no cytosol, some cytosol, or increased cytosolic volume, the latter created by laser-induced cell fusion followed by extrusion of one nucleus. Live imaging revealed that cell cycle timing was neither affected significantly by altering cell size nor by nuclear-to-cytosol ratio nor by DNA content [108]. Instead, cell fusion experiments revealed that cell cycle periods could be altered by the cytosol of a cell in a different cell cycle period. Together with other findings, these early results are consistent with studies that have posited models for the existence of “cytoplasmic factor(s)” that controls cell cycle timing [118,119,120]. The cytoplasmic factors that regulate many aspects of the cell cycle and its timing have now been identified [121]. 

Alterations of ploidy in embryos were also utilized to assess the scaling of the microtubule-based spindle [110]. Accurate chromosome partitioning relies on the assembly of a properly shaped spindle during cell division. The metaphase spindle forms a diamond shape, with two opposing corners at each centrosome and two at the ends of the metaphase plate (Figure 5E). The ratio between the spindle length (from the centrosome to centrosome) and width (of the metaphase plate at the equator) was constant throughout in vitro *Xenopus* extract studies that either molecularly or mechanically perturbed the dimensions of the metaphase spindles [122,123]. In vivo studies in *C. elegans* have been instrumental to our understanding of spindle shape. Initial studies showed that spindle length is affected by cell size as spindle length and width decrease and as embryogenesis progresses [110,118]. Interestingly, a reduction in the spindle width was observed when the spindle length was decreased by knocking down genes involved in spindle formation (e.g., by RNAi depletion of *tpxl-1* (the *C. elegans* homolog of *Xenopus* TPX2 involved in spindle assembly) or *spd-2* (involved in multiple aspects of centrosome biogenesis and thus spindle formation)) [124,125,126,127,128,129]. The opposite was not the case; changing the width of the spindle did not affect its length. In haploid embryos, the spindle width was reduced, and in polyploid embryos, it was increased [110]; yet in neither case was the embryo size or spindle length affected [124,125,126,127,128]. Correlation analysis between spindle length and width in haploid, diploid, and polyploid embryos revealed a simple equation that can predict spindle width based on the spindle length and ploidy of the embryo [110].

Another important correlation that was observed with the reduction in cell size during embryogenesis is related to chromosome condensation, a chromosome feature that is important for accurate chromosome inheritance [129,130,131,132,133]. Haploid and tetraploid embryos produced by diploid *klp-18*(*RNAi*) mothers show differences in the extent of chromosome condensation compared to diploid control embryos [130]. Chromosome condensation decreased in haploid embryos in comparison to diploid embryos, whereas condensation increased in tetraploid embryos. Differential condensation was independent of cell size, the speed of spindle elongation, and spindle length [110,130]. Therefore, an inverse correlation between nuclear DNA density and chromosome condensation was proposed and tested by observing chromosome condensation in dividing diploid and *klp-18*(*RNAi*) haploid embryos, during divisions as the size of the nucleus decreases during development [130]. The relationship between the DNA density in the nucleus and chromosome length (a measure of chromosome condensation) fits a regression line for the dividing diploid embryo. Importantly, this regression line also suitably fits the measurements of both the dividing haploid embryo and also embryos produced upon RNAi of factors that affect nuclear size including *ima-3* (nuclear pore import protein) and *ran-3* (ortholog of the human RCC1 regulator of chromosome condensation) [130]. These analyses revealed an allometric relationship between chromosome length/condensation and the relative ratio of DNA amount per nucleus size which is consistent with the hypothesis that chromosome condensation is regulated to adapt to fit the amount of DNA in the physical space within the nucleus.

## 4. Utilizing *Caenorhabditis* to Understand the Effects of Polyploidization

In addition to being utilized as tools to study development, *C. elegans* polyploids have also been used to study the process of polyploidization itself and its immediate after-effects.

### 4.1. Polyploid Tissues in C. elegans

At least two tissues are polyploid in *C. elegans*: the intestine and hypodermis (part of the nematode epidermis) (Figure 6) [86]. The intestine is a tube responsible for digestion and makes up nearly a third of the animal’s body. All intestinal cells are derived from a single blastomere (E) which is designated at the eight-cell embryo stage [134,135] (Figure 6A). E divides by mitosis, and then its progeny undergo rounds of endoduplication to allow the intestinal cells to massively increase nuclear output per unit volume which is thought to be necessary to meet the extensive enzymatic needs of this tissue.

The hypodermis has many roles in development including the establishment of the basic body plan in the embryo such as body shape and size, regulation of cell fate specification, and guidance of migrating axons and cells [136,137,138]. The hypodermal cell hyp7 is a multi-nucleated syncytium that encases most of the adult animal body (the tail and head of the animal are covered by smaller hypodermal cells) [86]. The hyp7 syncytium ultimately contains 139 nuclei in the adult which arise from cell fusions (Figure 6B) [86,139]. The embryonically derived nuclei in hyp7 remain as diploids, whereas ~98 cells generated post-embryonically endoduplicate to become tetraploids [86,87,139]. The ploidy of the hyp7 cell is key for adult animal size regulation, (see the section on allometric studies, below) [140].

In diploid cells, coordination of the centrosome and cell cycle ensures a single duplication of the centrosome per mitotic cycle, which is crucial to prevent supernumerary centrosome accumulation which can lead to genomic instability [140,141]. In cells that undergo endocycles, regulating the number of centrosome cycles is equally important for maintaining genome stability [142]. In the intestinal and hypodermal lineages of *C. elegans*, the centrioles become refractory to S-phase cues, and thus centriole duplication becomes uncoupled from the cell cycle, likely due to the loss of pericentriolar material, including the SPD-2 protein. After halting centriole duplication, these cells also eliminate all but one of the already duplicated centrioles [143]. Phosphorylation of the SPD-2 protein regulates centriole duplication; therefore, it represented a likely candidate for regulation of the centrioles during endocycles [144,145,146]. A CDK phospho-mutant, SPD-2^S545A^, prevents intestinal divisions due to a lack of centrosome duplication. In contrast, the phosphomimetic mutant, SPD-2^S545E^ leads to supernumerary centrosomes in the intestinal cells [143]. The effect of this allele is not observed in the hypodermis, suggesting cell-type-specific specializations in centriole control. SPD-2^S545E^ does not completely suppress the effects of *cdk-2*(*RNAi*), suggesting that additional phosphorylation sites are also involved. Indeed, SPD-2S^357^ was found to be regulated by Polo-Like Kinase-1 (PLK-1 in *C. elegans*), and a phosphomimetic mutation at this site resulted in the prolonged persistence of SPD-2 on centrioles and delayed centriole elimination in some of the intestinal nuclei. Surprisingly, PLK-1 does not seem to regulate either SPD-2 stabilization or centriole elimination [143] but appears to affect SPD-2 accumulation through its effect on the cell cycle [143]. Instead, knockdown of the *proteasome b-subunit* (*pbs-3*) results in both SPD-2 persistence indicative of a lack of centriole elimination in many of the intestinal cells. This finding implicates ubiquitin-mediated proteolysis in the uncoupling of the endocycle and centriole cycle in the intestine [143].

A central question in the WGD field has been whether there is a significant functional difference between mononucleated polyploidy (restricted to a single nucleus) and multinucleate polyploidy (in more than one nucleus) [2,147]. The developing intestine in *C. elegans* was used to address this question by converting the normally binucleated intestinal cells into mononucleated cells of identical ploidy using auxin-induced degradation [148] of either a regulator of mitotic entry (i.e., CDK-1) or a kinetochore protein required for chromosome partitioning in mitosis (i.e., KNL-1) to block endomitosis [38]. Whereas the absence of CDK-1 prevents entry into mitosis and essentially converts the endomitosis into an endocycle, degradation of KNL-1 allows for entry into mitosis but prevents the chromosome partitioning into two nuclei [38]. Knock-down affected neither cell size nor morphology, but the nuclear surface-to-volume ratio decreased in the mononucleated cells [38]. Single-worm transcriptome analyses revealed that the *vitellogenin* genes (*vit-1 to vit-6*) are downregulated in these mononucleated intestines. Vitellogenins are yolk proteins that are required for lipid transport to oocytes and embryos and are important determinants of progeny fitness. Accordingly, embryos from mothers with mononucleated intestines have reduced survival. Similarly, decreasing *vitellogenin* expression in binucleated gut cells phenocopies the impact on progeny fitness, and increasing *vitellogenin* expression, by overexpressing their transcriptional activators, results in mothers producing progeny with similar fitness regardless of whether they had mono- or bi-nucleated intestines [38]. Importantly, many of the genes that are downregulated in animals with mononucleated intestines normally increase in expression during the L4 to adult transition, implicating a major role in binucleation at this developmental stage [38].

### 4.2. Biological Size and Scaling (Allometry)

Regulation of cell size and scaling of tissues and organs are critical for development and influence homeostasis, metabolism, and function [33]. Polyploidization has been generally associated with increases in cell and organism size. For example, genome size and ploidy correlate linearly with cell volume in yeast [149,150,151]. However, the effects of polyploidy are cell-type specific in many multicellular organisms, with different cell types having a nonlinear relationship between ploidy and cell size [152]. For instance, in plants, there is a strong correlation between ploidy and cell volume in epidermal pavement cells, but the volume of palisade mesophyll cells remains at a constant size despite changes in ploidy [153]. These and other observations [154,155,156,157] make it apparent that the DNA mass does not fully explain the effect of ploidy on cell (and nuclear) size [158,159,160]. In most polyploidization models, including plants, interpreting the direct effect of ploidy is complex because the increase in cell size in tissues and organs is balanced by a reduction in cell number [33,157,160,161]. The invariant cell lineage in *C. elegans* simplifies the identification of mechanisms underpinning correlations between ploidy and scaling of cells, tissues, and organs.

Allometry studies related to the effects of ploidy in *C. elegans* have been performed for cells and animals [87,97,110,130,162,163,164,165,166]. Nigon (1949) [96] reported that compared to the 1300 µm-long diploid hermaphrodites, the 4A;3X hermaphrodites were on average 1360 µm and the 4A;4X hermaphrodites were 1560 µm long. The more profound increase in the 4A;4X hermaphrodites containing 24 chromosomes (See Figure 3) compared to the 4A;3X hermaphrodites (23 chromosomes) is surprising and suggests that gene expression from the X chromosome may contribute to an animal’s size significantly more than gene expression from autosomes.

The hypodermis secretes the cuticle forming the exoskeleton that houses the nematode’s internal organs and nervous system. There is a proportional relationship between the size of the *C. elegans* body and the size of the hypodermal cells (seam cells and the syncytium hyp7) [163,167,168]. Comparison of 12 nematode species from the order *Rhabditida* revealed a weak correlation between the number of nuclei and hypodermal volume (hyp7 in *C. elegans*), suggesting that body size was at least partly independent of cell number [87]. A significant correlation between animal size and the product of the number of nuclei and the ploidy was observed across species. Therefore, unlike most organisms where the evolution of body size relies on changes in cell number, in nematodes, body size evolution was likely driven by the size of the hypodermal syncytium [87]. A comparison of a tetraploid strain and its diploid revertant revealed a 39% increase in the adult body size of tetraploids [162]. The unexpected 1.4-fold increase in volume (instead of the expected 2-fold increase) in tetraploids is at least in part explained by the lower-than-expected ploidy in the hyp7 cell (16.7C instead of 22C) and the small reduction in the number of cells (by less than two nuclei) [162]. However, one should be careful in making conclusions from this comparison because it is likely that the reverted diploid differed from the original, but these were not examined. Multiple independent tetraploid strains from nearly any diploid strain provide the possibility of conducting this comparison more rigorously. Levels of endoduplication in hyp7 of *C. elegans* were found to be important in the regulation of the adult animal size. When *C. elegans* young adults were exposed to the DNA replication inhibitor hydroxyurea (HU), endoduplication of the hyp7 nuclei was inhibited, and a reduction in body size was observed [162]. Cyclin E is required for endoduplication in mice and *Drosophila* [169,170]. Null mutants carrying mutations of the *C. elegans* homolog of Cyclin E, *cye-1*, have an ~1.8-fold reduction in the size of the hyp7 syncytium and are 35–54% the size of wild-type worms [162].

Many signaling pathways and genes affecting diploid animal size have been characterized in *C. elegans*. These include pathways that only regulate embryo size, that regulate multiple developmental stages, and that influence adult body size. The TGFβ/BMP signaling pathway has a major role in the response to ploidy as it regulates both adult size and hyp7 ploidy [162,167,168,171,172,173]. The TGFβ signaling pathway that regulates body size in *C. elegans* includes the DAF-4 type-two receptor, the DBL-1 ligand, and the downstream SMADs (SMA-2, SMA-3, SMA-4, and SMA-6) [174]. DBL-1, the ortholog of human BMP7, is required for post-embryonic growth [87,166,175]. Whereas DBL-1 overexpression results in longer-than-normal animals (Lon phenotype) [172], mutations in the *dbl-1* gene cause a dwarf phenotype (Sma) and reduced hypodermal ploidy [162,172]. Interestingly, as with the *cye-1* mutant, exposure of *dbl-1* mutants to hydroxyurea neither further reduced hyp7 ploidy nor affected its body size, suggesting that DBL-1 normally promotes hyp7 endoduplication [162]. Together, these and other findings [162,168] suggest that DBL-1 regulates adult growth in a dose-dependent manner [165,172] by promoting endoduplication in the adult hypodermis (hyp7) [168], likely via CYE-1 cyclin [162].

The regulation of body size by the DBL-1 pathway is complex. Many of the molecular components in the DBL-1 pathway and other pathways affecting endoduplication in the hypodermis have been identified, and their localization and molecular or genetic interactions have been described. For instance, overexpression of LON-1, a member of the conserved PR-protein superfamily [164,176], causes hypo-endoduplication, and the absence of the LON-1 protein causes hyper-endoduplication in the hypodermis [164]. Specifically, the levels of the *lon-1* transcript depend on the dosage of the *dbl-1* gene and TGFβ signaling. Other genes and processes affecting diploid animal size are also likely to play important roles in this process. These include cuticle collagens, β-H Spectrin, TOR kinase, MAPK signaling, Hippo–Warts signaling, and the insulin signaling pathway [166,177,178,179,180,181,182,183,184,185,186]. These genes are expected to show altered expression in polyploid animals [33,41,187,188]. Global metabolic genes, such as rRNA or tRNA genes, are likely also involved in regulating biological scaling in polyploid animals [33,189]. Whether these pathways contribute equally to growth regulation in polyploids remains an open question.

## 5. Potential Future Queries Utilizing *C. elegans* Polyploids

The ease of creating synthetic autotetraploids from any strain utilizing RNAi is shepherding in a new era. We can now expand the use of *C. elegans* polyploids to study cellular and developmental processes (e.g., allometric relationships of different cell types and cell division dynamics) and disease (e.g., genome instability, pathological cell divisions, and cancer) by comparing equivalent organisms, tissues, organs, and cells from isogenic diploid, triploid, and tetraploid animals. This method allows for the first time in *C. elegans* the ability to query the direct impact of organismal polyploidization on individual tissues across time. In addition, this polyploid model’s unique features will be of particular benefit for advancing our understanding of WGD and its effect on genome structure, transcriptome size (total RNA per unit of DNA mass) [190], specific-gene expression, regulation of gene expression by chromatin modification, and importantly how these changes affect the physiology, metabolism, and developmental character of the organism.

In the course of evolution, parts of the genomes of polyploids are lost [191,192,193], a process known as fractionation [194]. This can occur because the purifying selection—the drive to remove mutated genes—is relaxed on duplicated genes, leading to both gene and regulatory element loss [194,195]. In allotetraploids, allele dominance can facilitate gene retention/fractionation, but genes involved in multi-subunit complexes (ribosomes, polymerases, etc.) tend to be retained [194]. Previous observations in plants showed that they undergo loss of repetitive sequences, mostly transposable elements, by recombination [187,188,189]. This can reduce the genome by up to 25% within the first few generations of newly formed autopolyploids [196]. Synthetic auto- and allo-tetrapolyploid *C. elegans* will allow both short-term and longitudinal studies of the process of genome downsizing given the nematodes’ short life cycle and their ability to manipulate their environment. Additionally, these animals would facilitate the study of tissue-specific differences and the mitotic stability of ploidy during development. Of particular interest will be comparative analyses of the endopolyploid hyp7 and intestinal cells to determine how their DNA size/mass scales with ploidy.

Gene expression studies have been performed to assess differential gene expression in allopolyploid plants, often comparing diploids to an ancestral hybrid species [33]. An important mystery remaining is the stoichiometry of the changes in whole transcriptomes of similar cells from animals with differing ploidy [190,197]. Two competing hypotheses regarding the effect of organism polyploidization on global gene expression have been proposed: (1) each duplicated genome retains diploid transcription levels or (2) all gene loci are affected by dosage compensation to maintain a gene dose similar to that of diploids. Although very few studies have been carried out to compare the effect of differing ploidy on transcriptome size, neither global suppression nor doubling of expression is uniquely likely since allopolyploid studies have shown an intermediate transcriptome size. Importantly, single-cell transcriptomic studies have shown high variability or noise in gene expression within a cell type and between different cell types. Specific regulatory genes (e.g., RNA polymerase 2 and 5.8S rRNA) have been found to differentially increase in expression with increasing ploidy [198]. Global and specific gene expression can also be affected by chromatin remodeling (e.g., chromatin and histone modifications), and specific histone modifications, particularly histone H3K56 acetylation, are thought to buffer increases in mRNA synthesis due to an increased number of gene copies [199].

The ability to create tetraploids with any strain facilitates longitudinal studies of the tetraploid. For example, one can now create recombinant inbred strains of auto- and allotetraploids to address how the genome changes over ten to hundreds of generations. Combining these studies with exposure to environmental stressors, such as disease, temperature, infection, and DNA damage, has the potential to shed light on genomic changes that provide flexibility and drive adaptation. A greater understanding of WGD and the mechanisms of tolerance to polyploidy will also have important implications for the treatment of cancer and other human pathogens which may gain a selective advantage over endogenous cells through these mechanisms.

## Figures and Tables

**Figure 1 jdb-11-00026-f001:**
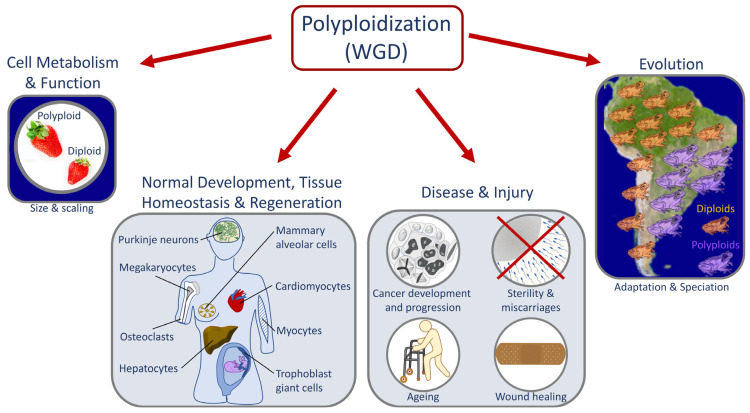
Roles of whole genome duplication (WGD). Diagram depicting key examples of processes affected by polyploidization/WGD. Polyploidy can occur at the cellular, tissue, or organismal levels. It is a key step in important biological processes affecting cellular metabolism and function, normal development, tissue homeostasis, regeneration, wound healing, organismal adaptation to the environment, and speciation. In addition, polyploidization can cause and/or prevent disease and repair injuries.

**Figure 2 jdb-11-00026-f002:**
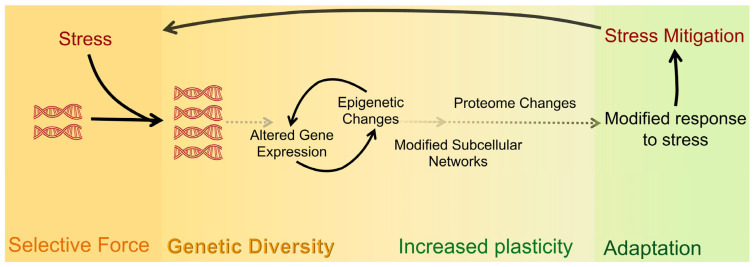
Shared causes and downstream effects of WGD across biological settings. Many forms of stress can lead an organism to undergo WGD. Whether as a response to the stress itself or as a way to adapt to the changing environment and/or different selective pressures, gene and epigenome changes lead to an altered proteome that supports an adaptive response to the initial stress exposure.

**Figure 3 jdb-11-00026-f003:**
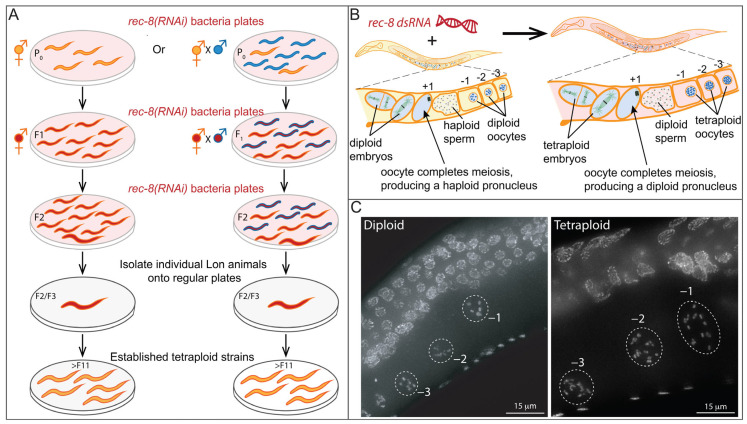
Method for rapid isolation of *C. elegans* tetraploids. (**A**). Diagram of the published method for generating tetraploid strains from diploids. This protocol relies on the transient knockdown of the meiosis-specific cohesin *rec-8* which results in the production of diploid instead of haploid gametes. Thus, upon fertilization, tetraploid progeny (which are longer and bigger) are produced. The method works both during the selfing of the hermaphrodites (**left**) and during outcrossing (**right**). Crossing allows for the generation of tetraploids with two different genetic backgrounds. After two to three generations of exposure to *rec-8* dsRNA, the larger hermaphrodites are individually plated on normal growth media. Strains that consistently produce large (Lon) hermaphrodites are then screened to confirm that these strains are tetraploids. (**B**). Diploid hermaphrodites (**left**) produce haploid gametes. Haploid sperm are stored in the spermatheca. Oocytes prior to the meiotic divisions have six pairs of connected homologous chromosomes that can be visualized by DAPI in the diakinesis-stage oocytes (Figure 3C, below) just adjacent to the spermatheca (−1, −2, and −3 oocytes based on position). The oocyte moves through the spermatheca and is fertilized and completes the meiotic divisions to produce a haploid pronucleus. The tetraploid worms derived from exposure to *rec-8* dsRNA, (**right**) produce diploid gametes with 12 DAPI bodies. (**C**) Examples of fixed hermaphrodites stained with DAPI to screen for ploidy by counting chromosome pairs in the −1, −2, and −3 diakinesis oocytes.

**Figure 4 jdb-11-00026-f004:**
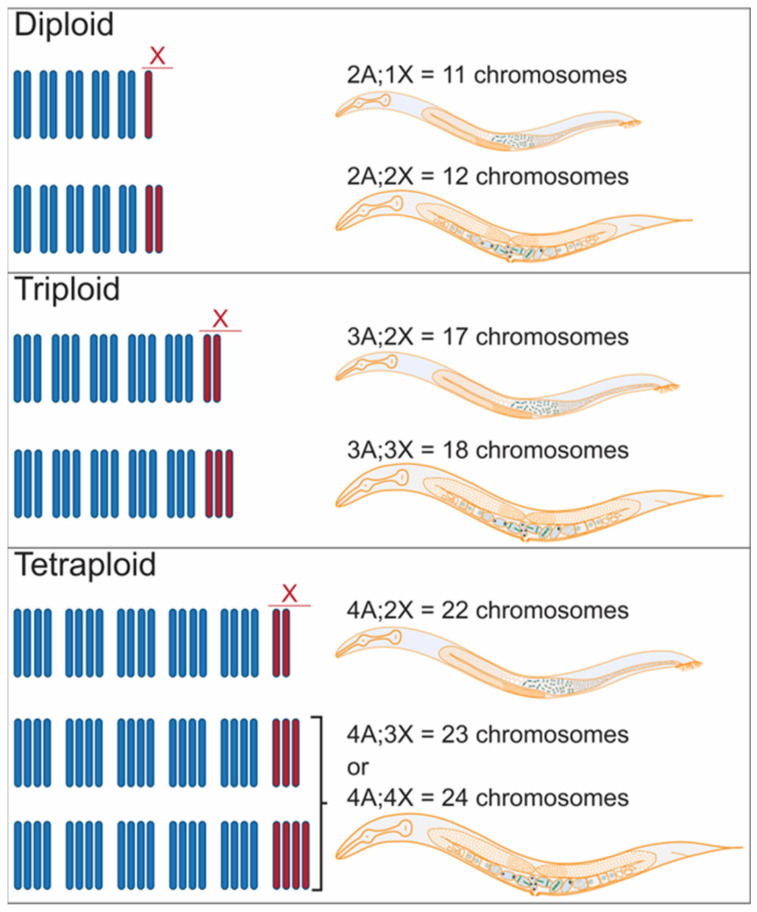
Sexual fate in *C. elegans* is determined by the X-chromosome/autosome ratio. *C. elegans* has six chromosomes, six autosomes, and one X chromosome. Diploid males and hermaphrodites have two sets of autosomes and one or two X chromosomes, respectively. Triploid males and hermaphrodites have 3 sets of autosomes and 2 or 3 X chromosomes making 17 or 18 total chromosomes, respectively. Tetraploid males have 22 chromosomes: 4 sets of autosomes (A) and 2 X chromosomes. In contrast, tetraploid hermaphrodites have four sets of autosomes and either three or four X chromosomes. See Table 1 for a summary of the X/A ratios that determine sex in *C. elegans*.

**Figure 6 jdb-11-00026-f006:**
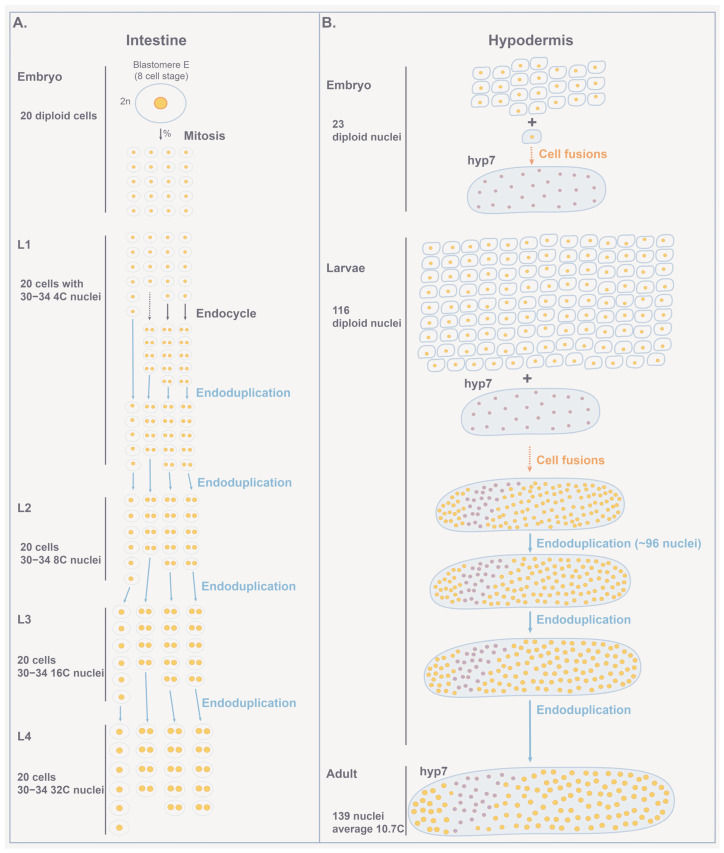
Polyploid tissues in *C. elegans*. (**A**) During embryogenesis, the E blastomere undergoes mitotic divisions to give rise to 20 intestinal diploid cells [86]. During the L1 stage, six of these cells neither divide nor replicate their DNA, four cells may or may not replicate their DNA or divide, and ten cells replicate their DNA and divide by endomitosis, producing cells with two diploid nuclei each. Therefore, by the end of the L1 stage, the 20 cells of the intestine have 30–34 diploid nuclei. Prior to the transition molt to the L2 stage, all nuclei of the intestine endoduplicate their DNA (4C each nucleus). Nuclei continue to endoduplicate prior to the molts between larval stages: the L2 (8C each nucleus), the L3 (16C), and finally the L4 (32C) larval stages. Thus, the adult intestine is composed of 20 cells with a total of 30–34 nuclei, each with 32C ploidy [86]. (**B**) Formation of the hypodermis. During embryogenesis, 23 cells fuse to the hyp7 cell. Post-embryonically, 116 cells fuse to the hyp7 syncytium. About 96 nuclei from these cells undergo one round of endoduplication. Most of these cells undergo two more rounds of endoduplication that take place post-embryonically to give rise to the adult hyp7 syncytium that contains 139 nuclei with 10.7C on average by the fully grown adult stage [87].

**Table 1 jdb-11-00026-t001:** X/A ratio determines sex.

Karyotype	X/A Ratio	Sexual Fate
2A;2X	1	female
2A;1X	0.5	male
3A;3X	1	female
3A;2X	0.67 *	Male
4A;4X	1	female
4A;3X	0.75 *	female
4A;2X	0.5	male

* X/A limits for female and male development. An X/A ratio ≥ 0.75 yields female fates and an X/A ratio ≤ 0.67 promotes male fates.

## Data Availability

Not applicable.

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
