# Peer review of "An Emerging Animal Model for Querying the Role of Whole Genome Duplication in Development, Evolution, and Disease"

_jdb, 2023, doi:10.3390/jdb11020026_

Round 1

Reviewer 1 Report

Review of:      An emerging animal model for querying the role of whole genome duplication in development, evolution, and disease

Summary:      Although most animals are diploid, there are rare species with more than two sets of chromosomes in the nucleus. Furthermore, even in diploid animals there are important tissues where cells are normally polyploid, and inappropriate polyploidization plays a role in many cancers. Here the authors present a lovely review of the broad field of polyploid studies, with a particular emphasis on using the roundworm C. elegans to address big questions about polyploidy in development, homeostasis and evolution. The writing is generally clear, the topic is very important, and the review is thorough and scholarly.

Recommendation:     I strongly recommend publication, after minor changes are made to improve the clarity of some passages.

Suggestions to clarify the discussion

(1) Around line 88, the authors mention polyploidy in some bacteria. Does the mechanism of bacterial DNA replication imply polyploidization is not very comparable to eukaryotes? I think the authors should expand on their inclusion of bacteria in this list, so that the reader has some idea of what is implied.

(2) In line 100, the authors write “or by expression changes that affect dose-dependent processes and complexes” when discussing lethality in polyploids. In this context it might be interesting to mention human triploids and tetraploids, and the likelihood that their severe symptoms are due to inability of expression from the single active X in each cell to match all of the extra sets of autosomes.

(3) Although the review is generally very clear, I found the part on centrosomes murky, and recommend that the authors spend some time polishing it.

(4) In lines 141-3, the authors mention the production of C. elegans males by errors in meiosis. There is a common misapprehension in the broader scientific community that this is the only way C. elegans males arise. It might be helpful to reword this passage to include the role of mating in producing males.

(5) Around line 161 the authors discuss trisomies. It is important to note here that the autosomal trisomies cause severe developmental problems, whereas the sex chromosome trisomies have only mild effects. (Probably because of Barr body formation).

(6) Around line 391, it might be nice to address the biological role of this endoreduplication.

(7) In lines 414-6 the authors describe the phosphomimetic mutant, SPD-2S545E. I found the summary confusing. They should describe clearly how the phosphomimetic mutation promotes centriole duplication, but does not completely suppress the effects of cdk-2 knockdown, suggesting that an additional phosphorylation site is also involved.

(8) In lines 439-40 the authors write: “ but the mononucleated intestinal cells have reduced nuclear to cytoplasmic ratio given that their nuclei more than double in size[49].” This doesn’t seem to make sense as written. If I understand the text, it would imply an increased ratio.

Minor editing changes

Line 15            Change “drugs resistance” to “drug resistance”

Line 54            Change “polyploidization is can cause” to “polyploidization can cause”

Line 108          Change “to addressing WGD” to “to studying WGD”

Line 121          Change “C. elegans”to “C. elegans

Line 125          Change “The method works for both during selfing” to “The method works both during selfing”

Line 137          Change “Caenorhabditis elegans” to “Caenorhabditis elegans

Line 147          “32C” I think this is the first place this notation is used. It should be defined here for the general reader.

Line 160          Change “except of the X chromosome” to “except of the X or Y chromosomes”

Lines 191-2     Change “About 66% of the induced tetraploids sired very low frequencies of males (0.6%), compared to the ~2% incidence of males in diploids” to About 66% of the induced tetraploids produced very low frequencies of males (0.6%), comparable to the ~0.2% incidence of males in diploids”

Line 192          Change “the remainder sired” to “the remainder produced” (It is best not to use ‘sired’ for production of progeny by self-fertile hermaphrodites, to avoid confusion).

Line 235          Change “involved in both” to “involved both in”

Line 241          Change “C. elegans” to “C. elegans

Line 356          By “show differential condensation of chromosomes” do the authors mean “show differences in the extent of chromosome condensation”? I am not sure what is being described here.

Line 366          Change “line also fitted well” to “line also fit well”

Line 366-370   This long sentence seems grammatically incomplete. Could it be rewritten for clarity?

Line 373          Change “to adapt to fit optimally the amount of DNA” to “to fit the amount of DNA”

Lines 382-3     Delete “The intestine of a newly hatched embryo has twenty diploid cells.”

Line 384          Change “Of these nuclei six do not divide nor replicate” to “Of these nuclei, six neither divide nor replicate”

Line 386          Change “divide by endomitosis cells with two diploid nuclei each” to “divide by endomitosis, producing cells with two diploid nuclei each”

Lines 405-6     Change “centrosomes accumulation” to “centrosome accumulation”

Lines 411-3     The acronym PCM is used only three times. It would be easier on the reader to write out “pericentriolar material” each time.

Line 414          Change “is key in for” to “is key for”

Lines 429-30   Change “whether there is a biologically-significant functional difference” to “whether there is a significant functional difference”

Line 443          Change “onto oocytes” to “into oocytes”

Line 484          Change “relies in changes” to “relies on changes”

Line 494          “in the hyp” Should this be hyp7?

Lines 507-8     Change “DBL-1, the human Ortholog” to “DBL-1, the ortholog”

Line 508          Change “for post-embryonic growth animal” to “for post-embryonic growth”

Line 517          Change “by DBL-1 pathway” to “by the DBL-1 pathway”

Lines 559-60   Change “gene expression between in allopolyploid plants” to “gene expression in allopolyploid plants”

Acknowledgements     Fix these to include actual acknowledgements.

Figure 2B        Something odd is happening to the text in this figure that needs to be fixed.

Figure 2C        The brightness should be increased using Photoshop so that the chromosomes are more easily visible

Figure 3           Why not shade the X chromosomes with a different color for clarity?

Table 1            For clarity, separate the diploids from the triploids with a space, and the triploids from the tetraploids with a space. Also, it might help to color the numbers with the same scheme as the words. Finally, use “hermaphrodite” rather than “female”

Author Response

Reviewer 1:

  1. Around line 88, the authors mention polyploidy in some bacteria. Does the mechanism of bacterial DNA replication imply polyploidization is not very comparable to eukaryotes? I think the authors should expand on their inclusion of bacteria in this list, so that the reader has some idea of what is implied.

We have added brief comments elaborating on the differential mechanism of bacteria polyploidization, and how regardless of this difference the effects of WGD in eukaryotes and prokaryotes are comparable.

Specifically, we added the following two sentences:

“In some cyanobacteria, polyploidy is established and maintained by misregulation of the replisome machinery[28–30].” And “In both prokaryotes and eukaryotes, the effects of WGD are comparable with increased resistance to UV irradiation, cell size, adaptability to environmental changes, and evolvability[28,31].”

(2)  In line 100, the authors write “or by expression changes that affect dose-dependent processes and complexes” when discussing lethality in polyploids. In this context it might be interesting to mention human triploids and tetraploids, and the likelihood that their severe symptoms are due to inability of expression from the single active X in each cell to match all of the extra sets of autosomes.

We have add the statement:  “The latter may be especially true in human triploid and tetraploids which may have difficulty equalizing expression between the silenced X chromosomes and the extra sets of autosomes, since variable numbers of Barr bodies may be found in these cells”

(3)   Although the review is generally very clear, I found the part on centrosomes murky, and recommend that the authors spend some time polishing it.

We have added a new figure and rewrote much of this section. We hope it is now clearer.

(4)   In lines 141-3, the authors mention the production of C. elegans males by errors in meiosis. There is a common misapprehension in the broader scientific community that this is the only way C. elegans males arise. It might be helpful to reword this passage to include the role of mating in producing males.

Thank you for this comment. We have added the statement: “Males are also produced by mating between hermaphroidtes and males (frequency >40%).” 

(5)  Around line 161 the authors discuss trisomies. It is important to note here that the autosomal trisomies cause severe developmental problems, whereas the sex chromosome trisomies have only mild effects. (Probably because of Barr body formation).

This now reads: “In humans, all monosomies, except of the X or Y chromosomes are lethal, whereas trisomies 13, 18, 21, X and Y are viable. The autosomal trisomies can cause severe developmental problems; sex chromosome trisomies have comparatively mild effects, presumably because of Barr body formation (extra X chromsomes) or the dearth of active genes (Y chromosome).”

(6)  Around line 391, it might be nice to address the biological role of this endoreduplication.

We have added the statement, “Endoreduplication allows cells to massively increase nuclear output per unit volume or to increase tissue mass.”

(7)  In lines 414-6 the authors describe the phosphomimetic mutant, SPD-2S545E. I found the summary confusing. They should describe clearly how the phosphomimetic mutation promotes centriole duplication, but does not completely suppress the effects of cdk-2 knockdown, suggesting that an additional phosphorylation site is also involved.

We have rewritten this section completely and hope it is now clearer to the reader. We also have included the relationship with cdk-2(RNAi) as an entrée to PLK-dependent phosphorylation and appreciate the reviewer pointing out this nuance.

8)    In lines 439-40 the authors write: “ but the mononucleated intestinal cells have reduced nuclear to cytoplasmic ratio given that their nuclei more than double in size[49].” This doesn’t seem to make sense as written. If I understand the text, it would imply an increased ratio.

We have modified the text to correct the error and now rephase to state that nuclear surface to volume ratio is decreased.

Minor editing changes:

We thank the reviewer for detailed reading of the manuscript and apologize for the textual errors. All of these have been corrected in the revised manuscript.

Fig 2B. Something odd is happening to the text in this figure that needs to be fixed.

We cannot see the problem with the text, but we replaced the figure in the hopes that this issue is corrected.

Fig 2C. The brightness should be increased using Photoshop so that the chromosomes are more easily visible.

We increased the brightness in photoshop of these images as requested.

Fig 3.  Why not shade the X chromosomes with a different color for clarity?

We colored in red the X chromosomes in the figure 3 for clarity.

Table 1 For clarity, separate the diploids from the triploids with a space, and the triploids from the tetraploids with a space. Also, it might help to color the numbers with the same scheme as the words. Finally, use “hermaphrodite” rather than “female”

We did this as well.

Reviewer 2 Report

The authors present a comprehensive overview of whole genome duplication at both the cellular and organismal and its role in evolutionary and disease processes. Further they highlight the challenges of developing polyploid laboratory models and deliver an in-depth history of Caenorhabditis elegans as a laboratory model of polyploidy and its use in investigating developmental processes. Finally, the authors describe the many potential research avenues that can utilize polyploid C. elegans to answer fundamental research questions.

Minor Comments

Line 191-192 – The authors write that Victor Nigon’s “induced tetraploids sired very low frequencies of males (0.6%), compared to the ~2% incidence of males in diploids” The incidence of males arising from nondisjunction is ~0.2% in diploids. Is this ~2% a typo?

-- Have there been any studies describing differences in tissue polyploidy between C. elegans isolates, such as N2 and CB4856? If so, the authors should include those papers in the review and discuss how tissue polyploidy may have arisen in those isolates and what function it may serve.

Author Response

Reviewer 2:

  1. Line 191-192 – The authors write that Victor Nigon’s “induced tetraploids sired very low frequencies of males (0.6%), compared to the ~2% incidence of males in diploids” The incidence of males arising from nondisjunction is ~0.2% in diploids. Is this ~2% a typo?

We are thankful to the reviewer their comments and for picking up the typo in pgs. 191-192. We have fixed the typo.

  1. Have there been any studies describing differences in tissue polyploidy between C. elegans isolates, such as N2 and CB4856? If so, the authors should include those papers in the review and discuss how tissue polyploidy may have arisen in those isolates and what function it may serve.

We are also thankful for the question about the different isolates of C. elegans ploidy. There are substantial copy number variation in coding sequences in the C. elegans genome in different isolates (e.g. Owen A. Thompson et al. 2015, Genetics 200(3): 975-989). These are unlikely to be due to WGDs.

  • There are differences between the N2 and Hawaiian strain, including significant differences in indel, but none of these could easily be explained by WGD.

  • There are no known wild-type polyploid strains of C. elegans or closely related species to our knowledge and we could not find reports of this in the literature either.

Reviewer 3 Report

In this Review the Authors discussed the significance of polyploidy in various biological contexts and in various biological scales.  Particular attention was focused on the importance of the model of polyploid C elegans. This is a good and comprehensive review where the Authors caught several very important issues that are still remaining almost unstudied.

 Specifically, the Authors outlined how C elegans model promotes understanding of the contribution of genome doublings (or polyploidy) to the regulation of meiotic and embryonic cell division and they uncovered the important connection between polyploidy and chromosome decondensation. The Authors also provided detailed description of polyploidy in various tissues of C Elegance and attracted attention to the complicated relationships between polyploidy and cell size. Specifically, they provided evidence that polyploidy does not always increase cell size in proportion to the number of genomes.

This review is a great contribution to the collection of facts that reveal the properties of polyploidy. It can be published. There are only several small recommendations:

Abstract:

1. Abstract is very good. It attracts interest and explains the importance of the study. Please, remove references and underline the most important and novel analytical findings of the Review.

2. If it is possible, in the first chapter explain why it is so important to study polyploidy. It would be good to mention that polyploidy plays an important role in gene regulation due to their epigenetic effects. Please, add more references concerning the relationships between polyploidy and chromatin spatial organization (for ex add DOI: 10.3390/ijms23179691 )

3. Just if it is possible, please, provide a small figure illustrating chapter 1.1. “Causes and downstream effects for WGD”

4. Please illustrate (or to provide a small table) for chapter 4.1.  “Polyploid tissues in C. elegans”  because the description of polyploidy in various tissues of C elegans is too technical. It also would be good to provide more evidence underlining the importance of the investigation polyploid C elegans for therapy of non communicative diseases, including cardiovascular diseases and cancer.

5. If it is possible, please, add more evidence confirming that polyploidy not always increase cell size proportionally to genome numbers (i.e. DOI: 10.1139/g04-015 )

6. In conclusion, please, outline the most important points of the Review and please, underline   applied and medical significance of polyploid C elegance model. 

There are only several small typos. The Authors will find them after re- reading.

PS It is a very good review. The chapter 4.1.  describing polyploidy in C elegans tissues is too technical and makes the Reader sleepy. It should be illustrated or changed :))

Author Response

Reviewer 3

  1. If possible, the first chapter should explain why it is important to study polyploidy. It would be good to mention that polyploidy plays an important role in gene regulation due to their epigenetics effects. Please, add more references concerning the relationships between polyploidy and chromatin spatial; organization. (DOI: 10.3390/ijms23179691)

This has been added to the text throughout.

  1. Just, if it is possible, please, provide a small figure illustrating chapter 1.1. “Causes and downstream effects for WGD”

We generated a new simplified figure (Figure 2) to address this suggestion.

  1. Please illustrate (or to provide small table) for chapter 3.1 “Polyploid tissues in C. elegans” because the description of polyploidy in various tissue in C. elegans is too technical. It also would be good to provide more evidence underlining the importance of the investigation polyploid C. elegans have for therapy of non-communicative disease, in CVD and cancer.

We have added a new figure (Figure 6) where the steps in polyploidization in the hypodermis and intestine have been added. This figure is significantly different from any available figure of these lineages in the consortia WormAtlas and WormBook publications.

  1. If it is possible, please, add more evidence confirming that polyploidy not always increase cell size proportionally to genome numbers. (10.1139/g04-015).

      We have added this reference to the appropriate section.

  1. In conclusion, please, outline the most important part of the review and please underline applied and medical significance of the polyploid elegans model.

We have added sentences throughout the text to address these comments. For instance, see in the last section of the manuscript.

Round 2

Reviewer 3 Report

The Authors addressed all recommendations and significantly improved the Review. They also added a lot of important information and prepared several great figures. I am sure that the Review will promote new studies and will attract citations. Our laboratory will cite it with great pleasure.

Congratulations with this wonderful work!